# Research and Application Progress of Resin-Based Composite Materials in the Electrical Insulation Field

**DOI:** 10.3390/ma16196394

**Published:** 2023-09-25

**Authors:** Bingyue Yan, Zhuo Zhang, Yin Li, Huize Cui, Chong Zhang, Jianfei He

**Affiliations:** State Grid Smart Grid Research Institute Co. Ltd., Beijing 102209, Chinayinli1@geiri.sgcc.com.cn (Y.L.);

**Keywords:** resin-based composite materials, molding process, insulation materials, insulation performance, mechanical properties

## Abstract

The research and application progress of resin-based composite materials in the field of electrical insulation has attracted considerable attention and emerged as a current research hotspot. This review comprehensively summarized the research and application progress of resin-based composite materials in the field of electrical insulation, providing detailed insights into their concept, properties, and preparation methods. In addition, a comprehensive evaluation of the electrical insulation performance, mechanical properties, and thermal properties of resin-based composite materials was presented, along with an in-depth analysis of their current application status. Despite the immense potential and development opportunities of resin-based composite materials, they also face several challenges. This review serves as a valuable reference and resource for researchers in related fields and aimed to promote further research and application development of resin-based composite materials in the field of electrical insulation.

## 1. Introduction

With the rapid development of science and technology, mankind’s demand and dependence on energy is growing. Against this background, it has become crucial to establish a reliable power system [1]. Power equipment such as cables, transformers, capacitors, and gas-insulated switches, as the core part of carrying, monitoring, and regulating the flow of electric power, are directly affected by the performance of their insulating materials. However, as power equipment gradually shifts from low-voltage, low-power, and large devices to high-voltage, high-power, and integrated development, traditional insulating materials (e.g., rubber, insulating varnish, and polymers) will not be able to satisfy the rapidly growing demand due to their shortcomings, such as poor mechanical properties, poor thermal conductivity, short service life, and degradation of electrical insulation properties, which are demonstrated at high voltages and temperatures. Therefore, the search for insulating materials with excellent mechanical properties, high thermal conductivity, and excellent electrical insulation has become the main research direction of researchers in this field.

To replace conventional insulating materials, composites have been extensively researched and developed because of their ability to combine the excellent properties of the internal component materials to improve the overall material performance. In insulating composites, polymers are considered to be the ideal matrix material because of their good electrical insulation properties, low cost, and excellent processability. Commonly used polymers include epoxy resins, phenolic resins (PF), polyether ether ketone (PEEK), polyphenylene sulfide (PPS), and polyetherimide (PEI), with resin materials being the most widely used [2,3]. However, resin materials have poor temperature resistance, low thermal conductivity, and weak mechanical properties, and, thus, cannot meet the demands of applications at high operating temperatures. In order to improve the thermal conductivity of resin materials, researchers have prepared various insulating resin matrix composites by adding fillers or fibers to the resin to improve its thermal conductivity. Common fillers include metals, ceramics, and carbon, while common fiber materials include glass fibers and carbon fibers. At present, there are already some insulating resin matrix composites that have been more maturely studied and applied, such as epoxy-resin-based glass fiber-reinforced composites, phenolic-resin-based glass fiber-reinforced composites, and epoxy-resin-based nanocomposites [4,5]. With the continuous pursuit of safety of power equipment, insulation performance, and temperature resistance in special environments, the research on resin matrix composites will continue to be in-depth and is expected to play a more important role in the field of insulation, providing lasting support for the reliable operation of power systems.

This paper focused on the progress of research and application of resin matrix composites in the field of insulation. The whole paper can be divided into the following parts: resin matrix composite system and its molding process, research on resin matrix composites in the field of insulation, and application of resin matrix composites in the field of insulation. Finally, the corresponding conclusions and references are given.

## 2. Resin Matrix Composite System and Its Molding Process

Resin matrix composites are a class of composites composed of resin matrix and reinforcement materials, which have the advantages of lightweight, high strength, excellent mechanical properties, and chemical resistance, etc. The resin matrix composites can be categorized into thermoset resin matrix composites and thermoplastic resin matrix composites. Depending on the resin matrix used, resin matrix composites can be divided into two categories: thermoset resin matrix composites and thermoplastic resin matrix composites. Among thermosetting resin matrix composites, the most widely used is epoxy resin matrix composites. In China, more than 10% of epoxy resins are used in composite manufacturing. In addition, phenolic resins are also widely used in resin materials for composites due to their excellent insulating properties, high thermal decomposition temperature, excellent flame-retardant properties, and low cost. In addition, specialty resins such as bismaleimide resins, phenolic resins, and cyanate ester resins are considered to be thermosetting resins with great potential for development due to their good electrical insulation, heat resistance, flame retardancy, transparency, and mechanical properties. Thermoplastic resin matrix composites often use general-purpose resins such as polypropylene (PP), polyethylene (PE), and polyvinyl chloride (PVC), as well as specialty resins such as polyphenylene sulfide (PPS) and polyether ether ketone (PEEK). Since thermoplastic resins are high molecular weight polymers, only a change in physical state occurs during processing, and unlike thermoset resins, they do not form a rigid, three-dimensional, cross-linked chemical structure, and, therefore, have the property of being repeatable [6,7].

Currently, there are several major molding processes for resin matrix composites, which are discussed below.

### 2.1. Resin Transfer Molding Process

The Resin Transfer Molding (RTM) molding process is one of the most practical processes available today. Figure 1 shows its basic principle: under specific temperature and pressure conditions, molten resin is injected into a mold with pre-positioned reinforcements such as fibers, and then cured and molded under appropriate temperature and pressure. The RTM process has the advantages of good molding results and applicability to a variety of shapes, but it also has some disadvantages, such as high cost and cumbersome process flow [8].

### 2.2. Compression Molding Process

Compression molding uses a similar principle to the RTM molding process. By using a hot press as a pressurized warming device, the prepreg is placed in a pre-designed mold, then the mold is closed and cured to shape. Alternatively, vacuum bags and hot press tanks can be used to achieve the same effect. The process has the advantages of simple process flow, fast molding speed, and flexible adjustment of molding parameters. As shown in Figure 2, a thermoplastic composite dot matrix structure prepared using a multiple molding process is demonstrated [9]. However, the design of the molds becomes complicated when preparing products with complex three-dimensional structures, and the need to redesign the molds when preparing products that are not mass-produced may lead to the problem of relatively low cost-effectiveness [10,11].

### 2.3. Automatic Fiber Placement In Situ Curing Process

The Automated Fiber Placement (AFP) in situ curing process is an innovative process for processing resin matrix composites and is shown schematically in Figure 3. The process combines two key steps, fiber placement and in situ curing, to enable the efficient, accurate, and automated production of composites. The basic principle of the AFP process is to use robots or automated systems to control the fiber placement device to stack prepregs layer by layer according to the design requirements while applying the appropriate pressure and heat sources to cure the resin. This process has many advantages such as high efficiency, high quality, and automation. However, it also suffers from higher costs and still faces some technical difficulties in the manufacture of large and complex structures and the automation of the process [12].

### 2.4. Hand-Paste Molding

Using manual work, the fiber fabric and resin are spread on the mold, bonded, and then cured into shape [13]. The process is simple and can meet the needs of processing complex structures, but it suffers from shortcomings such as low efficiency. In addition, the process is prone to generate a large amount of dust during operation, which can affect the health of the operators.

### 2.5. Wrap-Around Molding

The fibers are wound on the mold according to a certain pattern and then cured and demolded to form the product [14]. The process is highly reliable and has high production efficiency, and is widely used in the aerospace and military industry.

### 2.6. VARI Forming Process

VARI (Vacuum-Assisted Resin Injection) is a process that removes gas from fiber reinforcement in a vacuum state, impregnates fibers and their fabrics through resin flow and penetration, and solidifies at room temperature to form a certain fiber/resin ratio. This process method only requires vacuum pressure and no additional pressure. The entire operation process is carried out at room temperature without heating (depending on individual resin curing conditions, some require curing at medium to high temperatures). The resin fully penetrates the skin and sandwich structure under vacuum, greatly improving the overall structural strength and reducing defects and complex processes introduced by secondary bonding.

## 3. Research on Resin Matrix Composites in the Field of Insulation

### 3.1. Study of Electrical Insulation Properties

Electrical insulation properties are one of the most concerning properties in the research of insulating composites, which is also an important research hotspot in the development of electrical engineering. Insulating composites need to have the following electrical properties in electrical applications: (1) low conductivity; (2) high breakdown strength. The electrical insulating properties of composites are mainly determined by the electrical insulating properties of resin matrix. Epoxy resin composites are a class of insulating materials used in a large number of transmission cables, and their insulating properties are closely related to the safety of the transmission system. Organic compounds containing two or more epoxy groups in the molecule are called epoxy resins. This polymer material is generally presented as a liquid with a large viscosity at room temperature. In practical engineering applications, it is necessary to add different functional curing agents to the epoxy resin and then cure it at room temperature to form a solid polymer material with strong mechanical properties.

Epoxy resin composites have the advantages of good processing performance, good dielectric properties, and good stability, and, thus, are widely used in low, medium, high, and ultra-high voltage distribution networks and high-energy electrical devices. However, epoxy resin composites have the disadvantages of poor toughness, poor heat resistance, and poor moisture resistance; thus, their dielectric properties are significantly affected in complex application environments.

The electrical properties of epoxy resin composites are mainly evaluated by properties such as resistivity, dielectric constant, dielectric loss, and breakdown strength. Currently, the introduction of nanoparticles to prepare nanocomposite epoxy resin materials is the main way to enhance the electrical properties of epoxy resin composites. Aluminum trioxide nanoparticles are widely used as a common filler to optimize the electrical properties of epoxy resin composites. The research results [15,16] show that the properties of polymers can be regulated by controlling the content of nanoaluminum trioxide, as shown in the following: when a reasonable content of nanoaluminum trioxide is added, the breakdown strength of this composite will be significantly increased. When a low content of nano-aluminum trioxide is added (less than 1%), the dielectric constant as well as the dielectric loss of the nanocomposite will be lower than that of the resin composite.

Silica fillers also play an important role in the modification of epoxy composites. The results of the study show that the addition of nano-silica fillers in micron filler composite epoxy resin can improve the resistance of the material to partial discharge [17] and, at the same time, can reduce the accumulation of space charge after warming up. The addition of nano-silica particles in pure epoxy resin will first decrease and then increase the dielectric constant. In addition, the results [18] showed that the nano-silica fillers (mass fraction of 10%) can significantly promote the electrical properties of polymers, such as volume resistivity increased by 40%.

Some of the studies focused on the differences between the effects of different fillers. The results [16] show that nanoscale fillers have certain advantages over micrometer fillers when the filler content is small, as evidenced by the lower dielectric constant and dielectric loss of the nanoscale filler-blended epoxy composites. In addition, since nanoparticles only need to be much less than micron particles to bring about a larger interfacial area, with the increase in interfacial area, the impact strength and flexural strength of the epoxy resin material are improved. All of the above studies [19,20] show that various types of inorganic nanoparticle fillers have great potential for improving the electrical properties of epoxy composites.

In addition to inorganic oxide nanoparticles, conductive carbon nanotubes and metal nanoparticles can also be added to insulating polymers. When nanoconductor particles are uniformly dispersed in insulating polymers in a certain amount, they not only improve the breakdown strength of the composite material, but also maintain a good dielectric stability, which is due to the unique ‘Coulomb blocking’ effect of nanoparticles.

### 3.2. Electrical Corrosion Resistance Study

Epoxy resin composite materials will face various complex environments during service, and prolonged complex environments can lead to various electrochemical corrosion faults in the composite material, leading to electrical breakdown problems [21,22,23,24,25,26,27].

In high-voltage cables and cable accessories, the main electrical aging phenomenon that leads to insulation failure is electrical tree damage, a condition that leads to pre-breakdown of the insulation. At present, adding organic additives to the resin is one of the main methods to improve the electrical insulation properties of the resin. Organic additives mainly include antioxidants and ultraviolet absorbers. Antioxidants prevent the oxidative degradation of materials by inhibiting the generation of free radicals in the oxidative degradation process; ultraviolet absorbers can convert ultraviolet light into thermal radiation and higher wavelengths of radiation that are harmless to polymer-insulating materials. The addition of organic additives can improve charge modulation properties and reduce the probability of electrical tree damage. In addition, some researchers have developed self-healing insulating materials to address the problem of electric tree damage. For example, Figure 4 demonstrates polypropylene materials doped with functionalized nanoparticles, which are capable of repairing electric tree damage under specific conditions, which is important for improving the reliability of insulating materials [28,29,30,31].

In addition, space charge accumulation also has an important effect on the electrical insulation properties of materials. Even if the local electric field is not high, space charge accumulation will gradually degrade the dielectric properties and form defects, thus reducing the reliability and service life of the product [32]. Currently, there are three main methods to improve the inhibition of space charge accumulation in resin-insulating materials: (1) introduction of inorganic nanoparticles, (2) grafting of special functional groups, and (3) blending with other polymers. The current research on the above methods mainly focuses on improving the resin formulation. For example, previous research [33] demonstrates the distribution of space charge with time under 100 KV/mm DC field before and after the introduction of magnesium oxide (MgO) nanoparticles in polyethylene (PE). The results show that the introduction of nanoparticles can effectively inhibit the accumulation of space charge. In addition, researchers in the study of nano-ALOOH composite epoxy resin found that [33] the relative dielectric constant of the composite material increased with the introduction of nanoparticles, and nanoparticles can effectively improve the polymer’s ability to resist partial discharge.

In addition, the electrical insulation properties of resin matrix composites can also be studied by evaluating the breakdown strength and electrical strength to comprehensively understand and improve their properties, and the specific improvement means are the same as the above scheme. In summary, improving the electrical insulation properties of resin matrix composites mainly focuses on optimizing the formulation of the resin and using nanoparticles to improve its dielectric properties; however, it is difficult to satisfy other properties by only improving the electrical insulation properties of resin matrix composites, and there are major limitations in practical applications [34,35,36].

### 3.3. Thermal Conductivity Study

During the operation of power cables, the conductor temperature may reach 70 °C and 90 °C under both DC and AC conditions, while high voltage cables may even reach temperatures as high as 250 °C during an overload or short circuit. In practice, however, the heat generated by the center conductor of power cables accumulates on the inside of the insulation layer, and this heat is difficult to dissipate effectively due to the low thermal conductivity of the resin. As the transmission capacity and voltage level of the cable increase, the thickness of the insulating layer increases accordingly to cope with this problem, which further inhibits heat dissipation, leading to even more severe heat accumulation and seriously affecting the aging problem of the insulating equipment. Therefore, to improve the thermal conductivity of resins, researchers tend to add highly thermally conductive reinforcements to resins to prepare resin matrix composites with excellent thermal conductivity [37,38,39,40].

Epoxy resins, for example, have good thermal stability and low cost, but at the same time have low thermal conductivity (0.2 W/m-K) as well as a high coefficient of thermal expansion. To improve the thermal conductivity of epoxy resins, carbon, aluminum, magnetite, brass, copper, graphite, short carbon fibers, and copper nitride have been widely used as dopants to prepare epoxy resin-based composites. For example, Salunke et al. [41] investigated the effect of the size and content of BN in boron nitride (BN)-reinforced epoxy resin matrix composites on the thermal conductivity and dielectric properties of the composites. It was found that the thermal conductivity and electrical resistivity of the composites containing different sizes of BN could be effectively improved, and the relevant results are shown in Figure 5. In addition, Zhang et al. [42] doped BN into carbon-fiber-reinforced epoxy resin matrix composites, and similarly achieved the enhancement of the thermal conductivity of the composites. Donnay et al. [43] prepared boron nitride epoxy resin composites by the ball milling method, and explored the effect of micro- and nano-boron nitride particles on the thermal conductivity of the composites. The results showed that the thermal conductivity of boron nitride epoxy composites tended to increase linearly with the increase in the volume fraction of boron nitride. Huang et al. [44] chose spherical boron nitride and flaky boron nitride to investigate the effect of the shape of boron nitride on the thermal conductivity of the composites. The results show that the thermal conductivity of the composites shows an increasing trend with the increase in the volume fraction of spherical or flake boron nitride, but the flake boron nitride enhances the thermal conductivity of the composites more obviously. Spherical fillers can reach a large filling amount, but the contact area between the filler particles is relatively small, mainly in point contact, resulting in poor heat transfer efficiency of the material. Irregular-shaped fillers or sheet fillers have a larger contact area and better heat transfer effect, but the filling amount is smaller.

Carbon nanomaterials have extremely high thermal conductivity and can be utilized to prepare epoxy composites with high thermal conductivity [45]. The main types of carbon nanomaterials include graphene nanosheets, graphene oxide, and carbon nanotubes, etc. Wang et al. [46] investigated the thermal conductivity at the graphene and epoxy interfaces in graphene epoxy composites. It was shown that covalent and noncovalent functionalization could reduce the thermal resistance at the interface of the two materials. Oeksik et al. [47] used graphene-coated polymethylmethacrylate spheres as a filler to improve the thermal conductivity of epoxy composites. Shan et al. [48] successfully prepared graphene-oxide-reinforced epoxy nanocomposites. The results showed that graphene-reinforced epoxy nanocomposites have improved thermal conductivity while maintaining insulating properties.

Oxides have excellent mechanical properties and good insulating properties, which can also achieve the effect of enhancing the thermal conductivity of epoxy resin composites. Chen et al. [49] prepared epoxy resin materials modified with silica nanoparticles containing silver nanowires and found that the thermal conductivity of the modified resins increased with the increase in the volume fraction of silver nanowires when the mass fraction of silica was certain. Conversely, when the volume fraction of silver nanowires was certain, the thermal conductivity of the composites also increased with the increase in the volume fraction of silica.

### 3.4. Mechanical Properties

Resin matrix composites are usually subjected to severe static and dynamic loads when they are used as insulating materials, and the selection of lightweight and high-strength composites as insulating materials can effectively improve the safety of electrical equipment. At present, researchers have already made mature scientific achievements in the fields of wet thermodynamic properties of resin matrix composites, the influence of component material properties, and the influence of interfacial properties between components. According to the development of research methodology, in situ experiments at the micro- and nanoscale, macro-mechanical tests, and virtual tests based on multi-scale numerical simulations have been maturely applied to study the mechanical properties of resin matrix composites [50,51,52,53].

According to the different microstructures of the reinforcement in resin matrix composites, they can be divided into particle-reinforced resin matrix composites, continuous fiber-reinforced resin matrix composites, short-fiber-reinforced resin matrix composites, and fiber-cloth-reinforced resin matrix composites, etc. [54,55,56,57,58,59]. Jayaseelan et al. [60] investigated the mechanical properties of short-fiber-, microparticle-, and large-particle-reinforced epoxy resin matrix composites. The results showed that the tensile modulus of microparticle-reinforced epoxy resin matrix composites were optimal at 4.43 GPa, and the tensile strength and flexural modulus of short-fiber-reinforced epoxy resin matrix composites were the highest at 35.59 MPa and 2.41 GPa, respectively. In addition, a previous study [60] exhibits the impact strength values of the above composites, the results indicated that the impact properties of short fiber and large particle reinforced composites were the same, and the impact performance of micro-particles is poor.

Short-fiber-reinforced composites as well as particle-reinforced composites within the resin matrix as the main load-bearing structure leads to composite performance and matrix mechanical properties are the same, while continuous fiber-reinforced composites can effectively play the excellent mechanical properties of fibers and greatly enhance the mechanical properties of composites. As shown in Table 1, the mechanical properties of common resins, fibers, and fiber-reinforced resin matrix composites are demonstrated, and according to the data in the table, the mechanical properties of continuous fiber-reinforced resin matrix composites have been improved by at least one order of magnitude compared with those of resins.

## 4. Application of Resin Matrix Composites in the Field of Insulation

Resin matrix composites can be used in different forms as insulating materials in electrical equipment, microelectronics, LEDs, power generation, transportation, and aerospace, etc. [66,67,68,69,70]. As shown in Figure 6, resin matrix composites have been widely used in electrical and energy applications, where resin matrix composites are often used as the main material for insulating systems in high-voltage insulation, such as thermal conductive insulation systems for cables, generators, and transformers [23,71,72]. Resin matrix composites such as polyethylene-based nanocomposites and polyvinyl chloride composites are used for cable insulation; aramid fiber resin matrix composites are used for generator insulation systems; and fiberglass resin matrix composites are used for insulating equipment such as transformer-insulating cartridges [73]. Additionally, resin matrix composites are often used in the place of traditional ceramic materials as the main material for high-voltage insulators [74].

The above electrical equipment does not require high mechanical properties of insulating materials, but with the development of electrical equipment toward ultra-high voltage and other fields, breakthroughs in the mechanical properties of insulating materials have become an urgent problem to be solved at present. For example, the insulating tie rod as a circuit breaker transmission device plays the role of insulation and operation of the circuit breaker, which often needs to withstand a large switching overvoltage impact and compression, bending, stretching, and other mechanical loads. Therefore, it needs to have good resistance to bending, compression, and torsion, as well as good insulating properties. Currently, fiber-reinforced epoxy resin matrix composites have been widely used in the production of insulating tie rods, in which commonly used fibers include glass fibers, polyester fibers, and aramid fibers [75].

Composite materials have also been used to a certain extent in the field of power transmission towers. Traditional cement poles or metal transmission towers generally have problems such as high weight and short service life [76,77]. The use of composite materials to replace traditional transmission towers can effectively improve the level of lightning protection, and anti-fouling ability, and reduce line operation and maintenance costs. The U.S. Gar Wood Company developed composite poles as early as 1960, which were applied to the distribution network on the beaches of Hawaii, and the composite poles were subjected to complex environments such as high concentrations of salt spray and were still in service for decades [65]. In China, the relevant research work started later, Guangdong Power Grid Company began to study the composite tower technology in 2007 and researched on the electrical characteristics of the corresponding products, mechanical properties, aging resistance characteristics, and other key technologies [78].

Distribution network cross-arms are another major application of composites in insulation. Composite cross-arms are light in quality and high in strength, which can effectively reduce the difficulty of installation and improve construction efficiency [43]. The Pennsylvania Power and Light Industry Company in the United States applied the first batch of composite cross-arms from PUPI in 1990 [79]. The State Grid Corporation of China implemented a pilot project for the application of composite cross-arms in early 2016. The pilot found that composite crossarms can effectively improve the insulation level of lines and effectively prevent accidents such as ice-covered flashovers. Dubai in the Middle East pioneered the use of insulated crossarms on lines above 35 KV, achieving a significant reduction in line corridors [80,81].

Composite insulators have also been used in a large number of applications in power systems. Compared with traditional insulators, composite insulators are highly resistant to contamination, which significantly improves the safety and stability of the power system [82]. Europe and the United States were the first countries to use high-temperature vulcanized silicone rubber materials as insulating parts for electrical equipment around 1980 [83,84]. Currently, a large number of companies in the international arena have researched high-temperature vulcanized silicone rubber materials, such as the U.S. Dow Corning Corporation, the General Electric Corporation, and the Japanese Toshiba Corporation [85,86,87]. The composition of composite insulators consists of at least two insulating components [82,88], namely, the core rod and the umbrella sleeve. Composite insulators have been developed over the years to include the following forms [81,87]: rod suspension composite insulators, windproof composite insulators, pin insulators, cross-arm composite insulators, post composite insulators, composite interphase spacer rods, and composite insulators for electrified railroads.

As an important equipment for the construction of high-voltage power transmission projects, a high-voltage pipe sleeve plays the role of introducing or leading full voltage and current in the extra-high-voltage system and exerts its insulating and mechanical support effect. The composite high-voltage casing has been verified in the market for many years and has strong safety.

## 5. Conclusions

This paper introduced the research and application of resin matrix composites in the field of insulation, focusing on the preparation process, insulating properties, thermal properties, and mechanical properties of resin matrix composites, to reveal the direction of the research and development of current researchers. Based on the current research, it can be seen that the resin matrix composites already have a mature preparation process, gradually to the direction of intelligence and automation, and reduce the cost of preparation; in the research of the performance of resin matrix composites, focusing on the electrical insulation properties, thermal conductivity, mechanical properties, and other aspects. At present, the research focus of resin matrix composites in the field of insulation mainly tends to seek for resin matrix composites with excellent mechanical properties and good thermal conductivity and insulation properties at the same time. According to the current status of the application of resin matrix composites in the field of insulation, it can be seen that the resin matrix composites with excellent insulating properties will gradually replace the traditional insulating materials and become the main application in the field of insulation.

## Figures and Tables

**Figure 1 materials-16-06394-f001:**
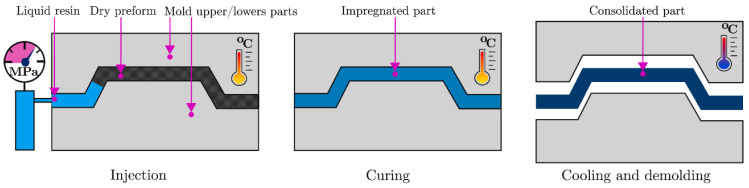
Schematic diagram of RTM molding process flow [8], (figures are reproduced and adapted with permission).

**Figure 2 materials-16-06394-f002:**
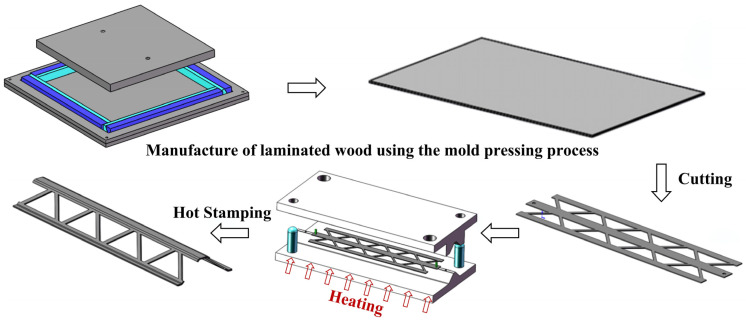
Structure of thermoplastic composite dot matrix prepared based on molding process [9], (Figures are reproduced and adapted with permission).

**Figure 3 materials-16-06394-f003:**
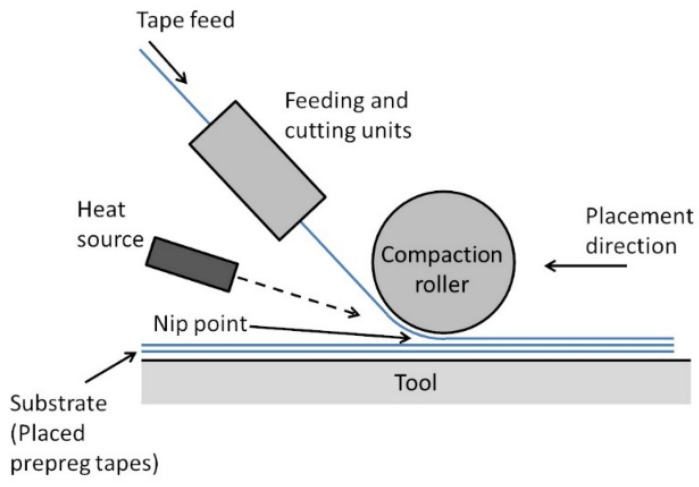
Schematic diagram of automatic fiber placement with in situ curing function [12], (figures are reproduced and adapted with permission).

**Figure 4 materials-16-06394-f004:**
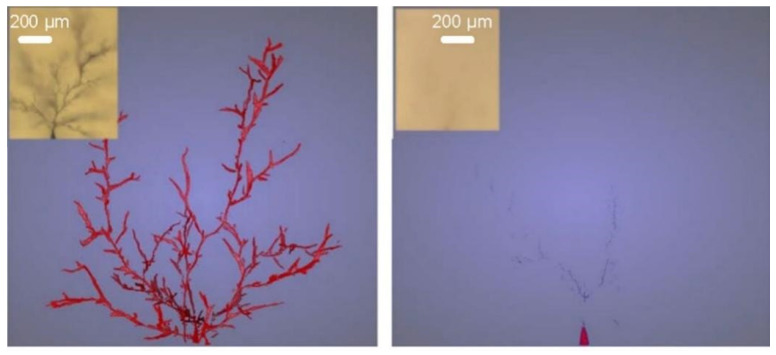
Electrical tree before and after PP repair doped with functionalized nanoparticles [28], (figures are reproduced and adapted with permission).

**Figure 5 materials-16-06394-f005:**
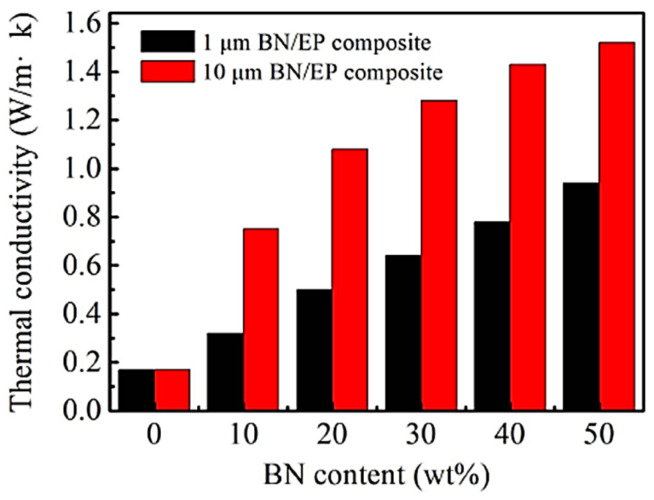
Thermal conductivity of reinforced composites with different BN sizes [41], (figures are reproduced and adapted with permission).

**Figure 6 materials-16-06394-f006:**
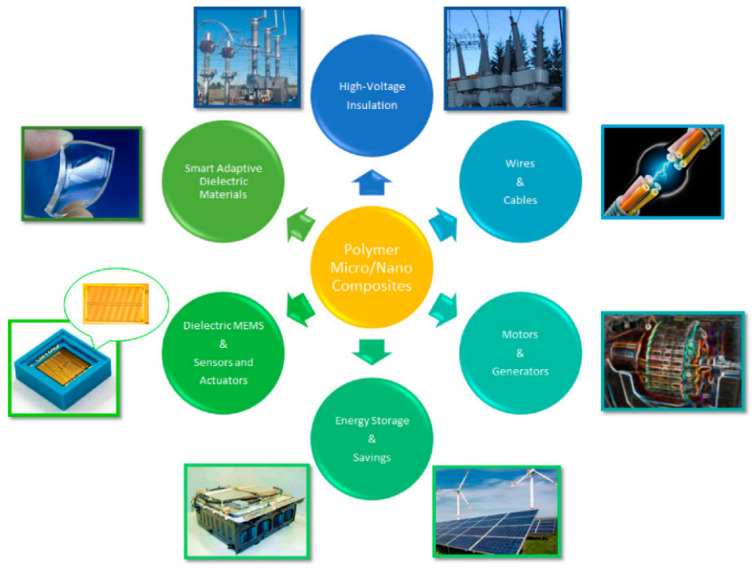
Application areas of resin matrix composites [23], (figures are reproduced and adapted with permission).

**Table 1 materials-16-06394-t001:** Basic mechanical properties of resins, fibers, and composites [61,62,63,64,65].

Material Type	Material Name	Density (g/cm^3^)	Modulus (GPa)	Tensile Strength (MPa)
Resin	Epoxy resin	1.1~1.3	3~4	60~95
	Phenolic resin	1.3	3.2	42~64
	Polyethylene	0.92	8.4	23
	Polypropylene	0.9	1.4	35~40
Fiber	Glass fiber E	2.55	74	3500
	Glass fiber S	2.49	84	4900
	Carbon fiber	1.75	225~670	750~7000
	Aramid fiber	1.47	134	2830
Composites	Glass/Epoxy	1.8	45	1370
	Carbon/Epoxy	1.5	155	1330
	Carbon/Polyetheretherketone	1.6	139	2280

## Data Availability

No new data were created or analyzed in this study. Data sharing is not applicable to this article.

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
