# Peer review of "Research and Application Progress of Resin-Based Composite Materials in the Electrical Insulation Field"

_materials, 2023, doi:10.3390/ma16196394_

Round 1

Reviewer 1 Report

Review of materials-2589617: Research and Application Progress of Resin-based Composite Materials in the Insulation Field.

Very interesting article. The authors describe the properties of general purpose and specialty resins. Methods of obtaining composites are also shown in the illustrations. The authors mainly included older, commonly known methods, and to a lesser extent, they discussed modern solutions. The influence of nanofillers on the properties of composites was also discussed. The resistance of composites to electrical corrosion, especially in high-voltage cables and cable accessories, to thermal and mechanical properties is discussed. The use of insulating materials in connection with their properties has been extensively discussed (point 4).

The authors quoted 89 literature items, most of them from the years 2019-22.

I my opinion the manuscript is suitable for printing, but I have a few comments.

1.     Verse 83 ,,one of the most mature’’ – can the word ,,mature’’ be replaced with another word? Mature is too general and doesn't capture the meaning of the sentence.

2.     To improve the value of the work, are the authors able to add any new, little-known methods of obtaining composites? The ones discussed here (page 3,4) are rather widely known. (partly mentioned in conclusion)

4.     Verse 180, the phrase "complex environments," is repeated: ,, Epoxy resin composites in service will face a variety of complex environments, complex environments for a long time will lead to all kinds of galvanic corrosion failure of composite materials, thus triggering the problem of electrical breakdown’’. Create two separate sentences separated by a period between these phrases.

5.     It would be useful to add mechanisms to improve various properties, e.g. the mechanism of action of boron nitride on the resin (verse 236-244) and others, e.g. the introduction of nanoparticles or oxides or spherical boron nitride and flaky boron nitride (which is the cause of the change in properties, e.g. why flake boron nitride enhances the thermal conductivity of the composites more obviously then spherical.

Reviewer 2 Report

It is an interesting review regarding resin based composites applications as electrical insulators. Both: production methods and main physical properties were followed. The manuscript evidences the materials aspects regarding the resin composites and various fillers that improve the polymer matrix behaviour. Overall it is a useful article for the specialists on the field. There are some aspects that require attention:

Comment 1) Title, Abstract – It is mandatory specifying that it is about Electrical Insulation

Comment 2) Line 169: The following sentence is not clear – the word ,,alpine” is not appropriate, please revise it ,, can significantly alpine the electrical properties”.

Comment 3) Line 173: You should expand the discussion of nano-filler efficiency compared to the micro filler particles for the electrical insulation effect.

Comment 4) It is important to discuss the composite electrical resistivity variation as a consequence of thermal conductivity enhancement with electrical conductive filler particles such as metals or carbon. The thermal conduction is enhanced but the insulator effect of the composite might decrease.

Comment 5) there are few issues regarding references that require proper revision:

Reference 14 is not complete. You give only an article identification number not a proper doi. Your reference [14] is most likely: Liyang Zhao, Susan C. Mantell, David Cohen, Reed McPeak, Finite element modeling of the filament winding process, Composite Structures, Volume 52, Issues 3–4, 2001, Pages 499-510, https://doi.org/10.1016/S0263-8223(01)00039-3.

References [33] and [34] are identical; therefore reference [34] must be replaced with another one.

Reference [35] is incomplete, it must be properly presented.

Reference [40] is published in 2021 not in 2019.

Reference [47] is published in 2007 not in 2010.

Reference [49] – volume number and pages missing, they must be added.

Reference [50] is not verifiable; therefore it must be replaced with another one with clear visibility.

Reference [79] is incomplete, it must be properly revised.

Moderate editing of English language required. There are some repeated sentences and some of the terms are confusing such as Line 169 ,, can significantly alpine the electrical properties”
